# Use of Laxative-Augmented Contrast Medium Increases the Accuracy in the Detection of Colorectal Neoplasms

**DOI:** 10.3390/diagnostics14171936

**Published:** 2024-09-02

**Authors:** Li-Yu Chen, Jong-Dar Chen, Yen-Kung Chen

**Affiliations:** 1Department of Family Medicine, Medicine and PET Center, Shin Kong Wu Ho-Su Memorial Hospital, No. 95, Wen Chang Rd., Shih Lin District, Taipei 111, Taiwan; leisurepika@gmail.com; 2School of Medicine, Fu Jen Catholic University, New Taipei City 242, Taiwan; 3Nuclear Medicine and PET Center, Shin Kong Wu Ho-Su Memorial Hospital, Taipei 111, Taiwan

**Keywords:** colon adenoma, colorectal cancer, laxative-augmented contrast medium, FDG PET/CT

## Abstract

Colonic adenomas are considered a precursor of colorectal cancer. A 75-year-old woman had a history of post-operation left breast cancer. She received an excision when the left chest wall recurred. A later FDG PET/CT scan revealed a focal intense FDG accumulation in the sigmoid, a focal mild FDG uptake in the pericolic lymph node, and a focal increased FDG accumulation in the transverse colon. A delayed FDG PET/CT scan after the per-rectal administration of the laxative-augmented contrast medium revealed a filling defect with persistent FDG uptake in the sigmoid and transverse colon and mild FDG uptake in the pericolic lymph node. In addition, more lesions were observed in the rectum and descending colon. The pathology reports showed sigmoid adenocarcinoma with lymph node metastasis, and adenomas in the transverse colon, descending colon, and rectum.

**Figure 1 diagnostics-14-01936-f001:**
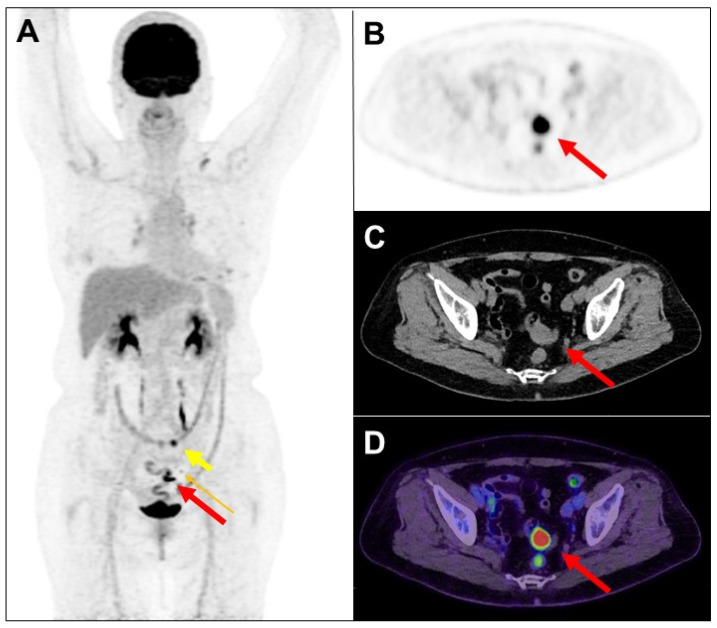
F-18 fluorodeoxyglucose (FDG) is excreted in part by the gastro-intestinal tract, with accumulation expected normally in the stomach and intestines. These physiologic sites of FDG accumulation may be confused with malignant lesions, and similarly increased FDG accumulation in malignant lesions may be interpreted as unrelated to cancer. A 75-year-old woman had a history of left breast adenoid cystic carcinoma following neoadjuvant chemotherapy and partial mastectomy. The pathology stage was pT2N0, and the Ki-67 index was 5%. Four years later, the patient’s left chest wall recurred and received excision. More than two weeks later, she received FDG positron emission tomography (PET)/computed tomography (CT) scan. The maximum-intensity projection view of PET image (**A**) revealed a focal intense FDG accumulation in the sigmoid (long thick arrow: red color), a focal mild FDG uptake in the pericolic lymph node (0.6 cm, long thin arrow: orange color), and a focal increased FDG accumulation in the transverse colon (short thick arrow: yellow color). Transaxial views of PET (**B**), CT (**C**), and PET/CT fusion (**D**) images showed a possible nodule in the sigmoid (long thick arrow: red color).

**Figure 2 diagnostics-14-01936-f002:**
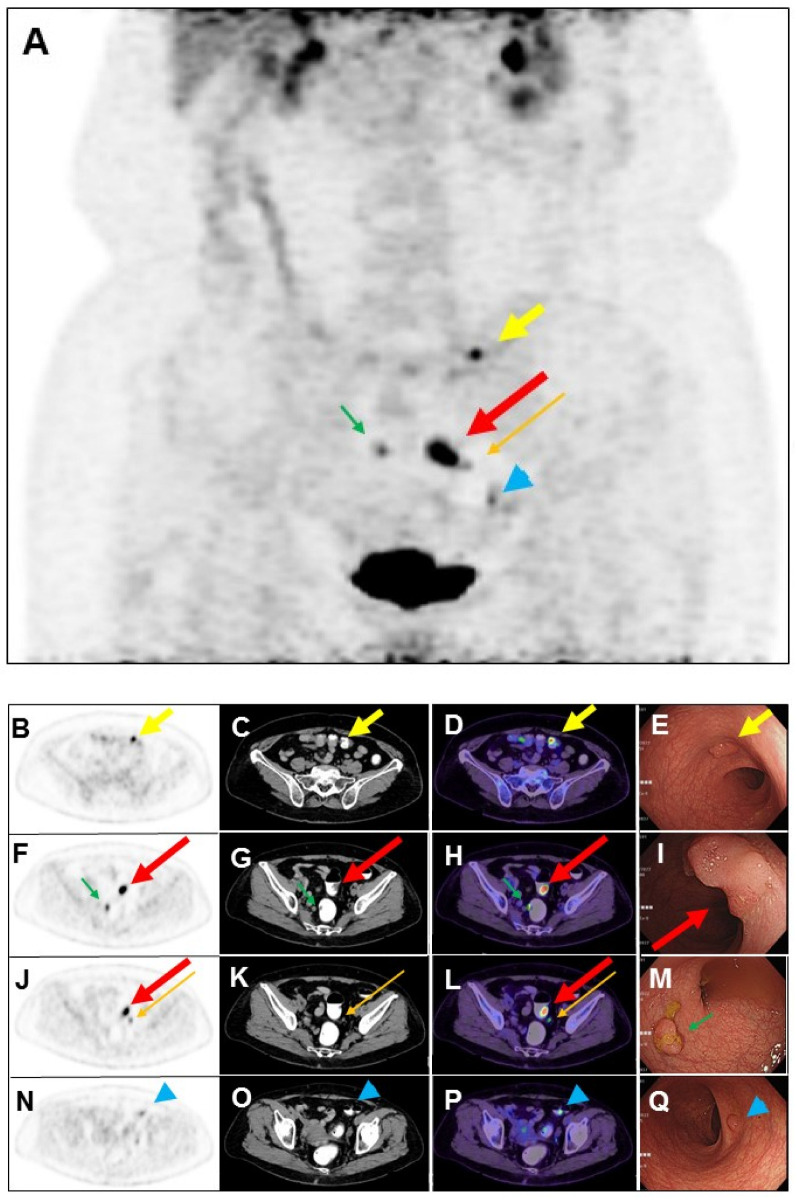
The use of a laxative-augmented contrast medium before a delayed FDG PET/CT scan leads to a reduction in the number of false-positive findings and increases the accuracy in the detection of colorectal cancer [1,2]. Maximum-intensity projection (**A**), transaxial PET, CT, and fused PET/CT images of delayed FDG PET/CT scan after per-rectal administration of laxative-augmented contrast medium revealed a filling defect with persistent intense FDG uptake in the sigmoid colon (1.8 × 1.8 × 0.6 cm; (**A**,**F**,**G**,**H**,**J**,**L**); long thick arrow), suggesting a hypermetabolic space-occupying lesion in bowel lumen. Further colonoscopy ((**I**), long thick arrow: red color) revealed a tumor in the sigmoid colon, which was proved to be adenocarcinoma. In addition, hypermetabolic nodule with filling defect in the transverse colon (1 × 0.8 × 0.5 cm; (**A**–**D**); short thick arrow) and rectum (1.3 × 0.8 × 0.6 cm; (**A**,**F**,**G**,**H**); short thin arrow: green color), colonoscopy, and histopathology revealed tubular adenoma and tubulovillous adenoma ((**E**), short thick arrow: yellow color; (**M**), short thin arrow: green color), respectively. Another focal mild hypermetabolic nodule in the descending colon (1 × 0.8 × 0.4 cm; (**A**,**N**,**O**,**P**); arrowhead: blue color) was also noted. The colonoscopy ((**Q**), arrowhead: blue color) and histopathology revealed tubulovillous adenoma. Later, the patient received laparoscopic anterior resection, and histopathology revealed moderately differentiated adenocarcinoma (2.0 × 2.0 × 0.6 cm) in stage pT1N1b. Lymph nodes had one metastasis to the pericolic ((**A**,**J**,**K**,**L**); long thin arrow: orange color) and inferior mesentery artery regions, respectively. Colorectal cancer is the third-leading cause of cancer death in the world [3]. Colorectal cancer usually begins with the most common form: an adenoma that originated from granular cells [4]. The proportion of colon adenoma to adenocarcinoma detected by FDG PET/CT scan is about 73% to 27% [5]. FDG PET is a sensitive imaging method for the detection of colorectal malignancy. However, benign, infectious, inflammatory, and granulomatous processes may also cause an increase in FDG uptake. Delayed FDG PET/CT performed after administration of a laxative-augmented contrast medium might be useful for identifying patients needing additional diagnostic procedures or avoiding unnecessary colonoscopic evaluation. The protocol consists of an initial phosphosoda enema (Fleet; C.B. Fleet, Lynchburg, VA, USA). Following the evacuation, 500 mL of diluted 3% contrast medium (Conray, iothalamate meglumine USP 60%; Mallinckrodt Inc., St Louis, MO, USA) is instilled into the anus. In this case, we used per-rectal administration of laxative-augmented contrast medium, and the detection of adenomas number was increased. An FDG-avid filling defect, suggesting hypermetabolic nodule with the space-occupying lesion, was discovered in an FDG PET/CT scan.

## Data Availability

Date available on request from the authors.

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
