# Peer review of "Use of Laxative-Augmented Contrast Medium Increases the Accuracy in the Detection of Colorectal Neoplasms"

_diagnostics, 2024, doi:10.3390/diagnostics14171936_

Round 1

Reviewer 1 Report (New Reviewer)

Comments and Suggestions for Authors

Although, the publication adds significantly to the published literature. However, there are some strong considerations to clear.

1. The authors have already published similar content, please clarify if the current work is as an invited commentary/short review/letter to the editor.

  • DOI: 10.1148/radiol.11101193
  • DOI: org/10.3390/diagnostics13132293

Author Response

Although, the publication adds significantly to the published literature. However, there are some strong considerations to clear.

  1. The authors have already published similar content, please clarify if the current work is as an invited commentary/short review/letter to the editor.

Ans: This is a rare case, delayed FDG PET/CT scan after per-rectal administration of laxative-augmented contrast medium revealed a filling defect with persistent FDG uptake in the sigmoid and transverse colon and mild FDG uptake in the pericolic lymph node. In addition, more lesions were observed in the rectum and descending colon.

Reviewer 2 Report (New Reviewer)

Comments and Suggestions for Authors

Clinical application of FDG PET/CT in diagnosis and TNM staging of cancer is well known. The use of a laxative as preparation for performing a FDG PET/CT has also been described in the literature for a long timeWe all known that use of laxative augmented contrast medium increases the accuracy in the detection of colorectal neoplasms.

1.   The authors have been using this technique for many years. If a protocol exists, it would be good to specify it

2.   Since the authors have already published numerous articles on the topic, perhaps it would be appropriate to start by highlighting what suggestions to provide to the scientific community. Examination with FDG PET/CT is very expensive, therefore cannot be included as a colorectal screening but as colorectal cancer is an extremely frequent tumor it could be useful to reflect on the opportunity to improve the accuracy of PET with the use of laxatives when used as staging for other types of tumors

3.   The references are appropriate.

4.   Tables and figures are appropriate.

Author Response

Clinical application of FDG PET/CT in diagnosis and TNM staging of cancer is well known. The use of a laxative as preparation for performing a FDG PET/CT has also been described in the literature for a long time. We all known that use of laxative augmented contrast medium increases the accuracy in the detection of colorectal neoplasms.

Ans: Before, delayed FDG PET/CT scan after per-rectal administration of laxative-augmented contrast medium revealed marked decreased false positive, so increased accuracy. In this case, more lesions were observed in the rectum and descending colon. So, sensitivity increased, accuracy increased.

  1. The authors have been using this technique for many years. If a protocol exists, it would be good to specify it

Ans: This case is accident findings, more lesions of adenoma.

  1. Since the authors have already published numerous articles on the topic, perhaps it would be appropriate to start by highlighting what suggestions to provide to the scientific community. Examination with FDG PET/CT is very expensive, therefore cannot be included as a colorectal screening but as colorectal cancer is an extremely frequent tumor it could be useful to reflect on the opportunity to improve the accuracy of PET with the use of laxatives when used as staging for other types of tumors

Ans: Yes.

  1. The references are appropriate.

Thanks.

  1. Tables and figures are appropriate.

Thanks.

Reviewer 3 Report (New Reviewer)

Comments and Suggestions for Authors

I appreciate the opportunity to review this article. It is an "interesting image" article on the use of laxative-enhanced contrast media to enhance the detection of colorectal neoplasms. The images presented are clear and well described; however, I feel that the article in its current form has several details that should be corrected before publication is considered:

-The article lacks an introduction and description of the case, and in general it lacks a clear structure. In Figure 1, the case appears to be described, but the figure and its description are placed first, so it is not clear if the entire description corresponds to this page or if it is worth restructuring it as a description of the case before presenting the figure(s). -In Figure 2, several letters are described (which, it seems, correspond to the images added as optional in a separate file, and therefore they do not seem to describe the image in the body of the article). In addition, a description of the method for creating the contrasting images is placed in the same description of that image, but it is not related to the description and should be placed as an introduction. -The title of the article makes a speculative statement (it claims to increase the accuracy of detection) that cannot be substantiated by the presentation of a single case. In addition, a statement is added between lines 67-69 that is not supported by the information presented.

Author Response

I appreciate the opportunity to review this article. It is an "interesting image" article on the use of laxative-enhanced contrast media to enhance the detection of colorectal neoplasms. The images presented are clear and well described; however, I feel that the article in its current form has several details that should be corrected before publication is considered:

-The article lacks an introduction and description of the case, and in general it lacks a clear structure. In Figure 1, the case appears to be described, but the figure and its description are placed first, so it is not clear if the entire description corresponds to this page or if it is worth restructuring it as a description of the case before presenting the figure(s). -In Figure 2, several letters are described (which, it seems, correspond to the images added as optional in a separate file, and therefore they do not seem to describe the image in the body of the article). In addition, a description of the method for creating the contrasting images is placed in the same description of that image, but it is not related to the description and should be placed as an introduction. -The title of the article makes a speculative statement (it claims to increase the accuracy of detection) that cannot be substantiated by the presentation of a single case. In addition, a statement is added between lines 67-69 that is not supported by the information presented.

Response: An introduction in figure 1: F-18 fluorodeoxyglucose (FDG) is excreted in part by the gastro-intestinal tract, with accumulation expected normally in the stomach and intestines. These physiologic sites of FDG accumulation may be confused with malignant lesions, and similarly increased FDG accumulation in malignant lesions may be interpret as unrelated to cancer.

An introduction in figure 2: The use of a laxative-augmented contrast medium before a delayed FDG PET/CT scan leads to a reduction in the number of false-positive findings and increases the accuracy in the detection of colorectal cancer.

Round 2

Reviewer 2 Report (New Reviewer)

Comments and Suggestions for Authors

The revisions made are adequate and the manuscript is deemed acceptable

Author Response

Clinical application of FDG PET/CT in diagnosis and TNM staging of cancer is well known. The use of a laxative as preparation for performing a FDG PET/CT has also been described in the literature for a long time. We all known that use of laxative augmented contrast medium increases the accuracy in the detection of colorectal neoplasms.

Response: Before, delayed FDG PET/CT scan after per-rectal administration of laxative-augmented contrast medium revealed marked decreased false positive, so increased accuracy. In this case, more lesions were observed in the rectum and descending colon. So, sensitivity increased, accuracy increased.

  1. The authors have been using this technique for many years. If a protocol exists, it would be good to specify it

Response: This case is accident findings, more lesions of adenoma.

  1. Since the authors have already published numerous articles on the topic, perhaps it would be appropriate to start by highlighting what suggestions to provide to the scientific community. Examination with FDG PET/CT is very expensive, therefore cannot be included as a colorectal screening but as colorectal cancer is an extremely frequent tumor it could be useful to reflect on the opportunity to improve the accuracy of PET with the use of laxatives when used as staging for other types of tumors

Response: Yes, if images revealed focal FDG uptake in the colon.

Reviewer 3 Report (New Reviewer)

Comments and Suggestions for Authors

The authors have responded to most of my comments, the presentation of the images is clearer, and the information presented adds more context to highlight the importance of the images.

As a single comment, I continue to insist, as I mentioned in the last review, that the title of the article should be slightly modified; instead of saying "increases", it should say "may increase" or "increased", this so that the title does not imply that this finding presented with a single case is sufficient evidence to support this statement in a general way.

Author Response

Response: An introduction in figure 1: F-18 fluorodeoxyglucose (FDG) is excreted in part by the gastro-intestinal tract, with accumulation expected normally in the stomach and intestines. These physiologic sites of FDG accumulation may be confused with malignant lesions, and similarly increased FDG accumulation in malignant lesions may be interpret as unrelated to cancer.

An introduction in figure 2: The use of a laxative-augmented contrast medium before a delayed FDG PET/CT scan leads to a reduction in the number of false-positive findings and increases the accuracy in the detection of colorectal cancer.

This manuscript is a resubmission of an earlier submission. The following is a list of the peer review reports and author responses from that submission.

Round 1

Reviewer 1 Report

Comments and Suggestions for Authors

The present paper by Li-Yu Chen, MD, PhD et al. described the report, “Use of laxative-augmented contrast medium increase the accuracy in the detection of colorectal neoplasms”. In this report, they demonstrated how to use laxative-augmented contrast medium in PET/CT. However, it seems somewhat less novel than previous report.

Author Response

Comments 1:

The present paper by Li-Yu Chen, MD, PhD et al. described the report, “Use of laxative-augmented contrast medium increase the accuracy in the detection of colorectal neoplasms”. In this report, they demonstrated how to use laxative-augmented contrast medium in PET/CT. However, it seems somewhat less novel than previous report.

Answer: We used per-rectal administration of laxative-augmented contrast medium, and the detection of adenomas number was increased.

Reviewer 2 Report

Comments and Suggestions for Authors

This case report presents a case it addresses the persistent challenge of distinguishing colonic adenomas and colorectal cancer. In essence, the report offers a comprehensive examination of a 75-year-old woman had a history of left breast cancer and left chest wall recurrence who underwent FDG PET/CT. Upon the completion of essential revisions, I believe this report holds potential for acceptance in a scholarly publication.

The following recommendations are proposed:

1.It is recommended that the author provide a detailed background and appropriate discussion of the case.

2.Could the author explain the advantages of FDG PET/CT compared to colonoscopy? 

3.In relation to Figure 2, if feasible, the author is advised to enhance the clarity of the image.

4.The presence of grammatical errors has been identified. It is suggested that the author review and rectify them accordingly.

5.It is suggested that the author make some adjustments to the format in accordance with the standard requirements of the journal.

Comments on the Quality of English Language

Moderate editing of English language required.

Author Response

Comments 2:

The following recommendations are proposed:

  1. It is recommended that the author provide a detailed background and appropriate discussion of the case.

Answer: FDG is excreted in part by the gastro-intestinal tract, with accumulation expected normally in the stomach and intestines. These physiologic sites of FDG accumulation may be confused with malignant lesions, and similarly increased FDG accumulation in malignant lesions may be interpret as unrelated to cancer.

The use of a laxative-augmented contrast medium before a delayed FDG PET/CT scan leads to a reduction in the number of false-positive findings and increases the accuracy in the detection of colorectal cancer.

  1. Could the author explain the advantages of FDG PET/CT compared to colonoscopy?

Answer: Do not needs anesthesia.

Detected other tumors (lung cancer, breast cancer, lymphoma, etc.), except for colorectal tumors. 

  1. In relation to Figure 2, if feasible, the author is advised to enhance the clarity of the image.

Answer: OK.

  1. The presence of grammatical errors has been identified. It is suggested that the author review and rectify them accordingly.

Answer: OK.

  1. It is suggested that the author make some adjustments to the format in accordance with the standard requirements of the journal.

Answer: OK.
